# Candidemia in Children with Malignancies: Report from the Infection Working Group of the Hellenic Society of Pediatric Hematology-Oncology

**DOI:** 10.3390/jof6040276

**Published:** 2020-11-10

**Authors:** Eleni Vasileiou, Anna Paisiou, Charoula Tsipou, Apostolos Pourtsidis, Vasiliki Galani, Nikolaos Katzilakis, Kondilia Antoniadi, Eugenia Papakonstantinou, Elda Ioannidou, Efthichia Stiakaki, Margarita Baka, Antonios Kattamis, Vasiliki Kitra, Athanasios Tragiannidis

**Affiliations:** 1Hematology-Oncology Unit, 2nd Department of Pediatrics, Aristotle University of Thessaloniki, AHEPA Hospital, 546 36 Thessaloniki, Greece; eleni.vasileiou@yahoo.com; 2Stem Cell Transplant Unit, Aghia Sophia Children’s Hospital, 115 27 Athens, Greece; salamanca.creta@yahoo.gr (A.P.); elda.ioannidou@gmail.com (E.I.); vkitra@hotmail.co.uk (V.K.); 3Pediatric Hematology-Oncology Unit, 1st Department of Pediatrics, National and Kapodistrian University of Athens, Aghia Sophia Children’s Hospital, 115 27 Athens, Greece; charoula.tsipou@gmail.com (C.T.); ankatt@med.uoa.gr (A.K.); 4Oncology Department, P. and A. Kyriakou Children’s Hospital, 115 27 Athens, Greece; tolispou@gmail.com (A.P.); margbaka@hotmail.com (M.B.); 5Pediatric Hematology Oncology Department, Mitera Children’s Hospital, 151 23 Athens, Greece; vicky_gln@hotmail.com; 6Pediatric Hematology Oncology Department, University of Crete, 700 13 Heraklion, Greece; katzilaher@yahoo.gr (N.K.); efstel@med.uoc.gr (E.S.); 7Department of Pediatric Hematology-Oncology, Aghia Sophia Children’s Hospital, 115 27 Athens, Greece; condilia.and@gmail.com; 8Department of Pediatric Hematology and Oncology, Hippokration Hospital, 546 42 Thessaloniki, Greece; eugepapa@yahoo.gr

**Keywords:** candida, candidemia, hematology, oncology, pediatric, epidemiology, risk factors, treatment, *C. parapsilosis*

## Abstract

Candidemia is an important cause of morbidity and mortality especially in immunocompromised and hospitalized patients. We retrospectively collected data of candidemia cases that occurred in the seven Hematology-Oncology Departments/Units of Greece and the Stem Cell Transplant Unit between 2015 and 2019. In total, 19 episodes of candidemia in 19 patients were recorded. The majority of the patients (78.9%) had at least one risk factor for candidemia. The most frequent risk factors associated with candidemia observed in our patients were prolonged duration of hospitalization (30 days, range 1–141), presence of a central venous catheter at diagnosis of candidemia (73.7%) and antibiotics use during the last two weeks (84.2%). *Candida parapsilosis* was the most common species isolated accounting for 42.1%, followed by *C. albicans* (26.3%) and *C. famata* (15.8%). Nearly all of the patients (84.2%) received antifungal monotherapy with liposomal amphotericin B or echinocandins. The central venous catheter was removed in 78.6% of patients and the median time between the first positive blood culture and catheter removal was 3 days (range 1–9). Mortality at 28 days was 26.3%. In conclusion, a predominance of non-albicans species was observed in our study in conformity with the global trend.

## 1. Introduction

Candidemia and invasive candidiasis are important causes of morbidity and mortality mainly in immunocompromised and hospitalized patients associated with high financial burden [1,2,3]. The annual incidence rate of candidemia is estimated to be 3.88 per 100,000 inhabitants, while reported incidence rates range from 1.0 to 10.4 [4]. Moreover, candidemia is the leading cause of invasive fungal infections in hospitalized children, particularly those undergoing intensive chemotherapy for acute leukemia and heterologous stem cell transplantation [5]. There are approximately two hundreds of Candida species, but those most commonly associated with invasive disease in humans are *C. albicans*, *C. tropicalis*, *C. parapsilosis*, *C. glabrata*, *C. krusei*, and *C. auris* [6]. Although, it is widely accepted that the intestinal tract is the main source of candidemia, the importance of the colonization of the skin by *C. parapsilosis* has recently become apparent [7,8]. Major risk factors associated with candidemia are age < 1 year or >65 years, surgery, presence of a central venous catheter, total parenteral nutrition, previous antibiotic exposure, azotaemia, graft-versus-host disease and neutropenia after induction chemotherapy in acute leukemia patients [9,10,11,12]. Echinocandins and liposomal amphotericin B (L-AmB) are the first-line antifungal agents in the treatment of candidemia both for immunocompetent and immunocompromised patients and at the same time central venous catheter (CVC) removal is strongly recommended according to international consensus guidelines [13,14,15,16]. The mortality rate of candidemia at 30 days for children in Europe ranges between 1.8% and 27.8%. Higher mortality rates are also observed for *C. tropicalis* (21.3%) and *C. krusei* (19.3%) compared to *C. albicans* (13.6%) and *C. parapsilosis* (12.7%) [17].

The aim of our study was to report the cases of candidemia in pediatric patients with malignancies in seven Hematology-Oncology Departments and the Stem Cell Transplant Unit in Greece during the years 2015–2019.

## 2. Materials and Methods

The Infection Working Group (IWG) of the Hellenic Society of Pediatric Hematology-Oncology (HELSPHO) collected and analyzed all cases of candidemia that occurred in the seven Hematology-Oncology Departments/Units of Greece (four in Athens, two in Thessaloniki, one in Crete) and the Stem Cell Transplant Unit (Athens, Greece). In particular, the Pediatric Hematology-Oncology Unit of the 1st Department of Pediatrics and the Department of Pediatric Hematology-Oncology of Aghia Sophia Children’s Hospital in Athens, the Oncology Department of the P. and A. Kyriakou Children’s Hospital in Athens, the Pediatric Hematology Oncology Department of the Mitera Children’s Hospital in Athens, the Hematology-Oncology Unit of the 2nd Department of Pediatrics in AHEPA Hospital in Thessaloniki, the Department of Pediatric Hematology and Oncology of Hippokration Hospital in Thessaloniki, the Pediatric Hematology Oncology Department of General University Hospital of Heraklion in Crete and the Stem Cell Transplant Unit of Aghia Sophia Children’s Hospital in Athens. Our retrospective observational study was conducted from January 2015 to December 2019. Candidemia was defined as the presence of any *Candida* species in the blood and the date of diagnosis corresponded to the day of the first positive blood culture. Data were collected through a review of the microbiology and clinical records. For each patient demographic data, underlying diseases and comorbidities, clinical characteristics (neutropenia defined as an absolute neutrophil count < 0.5 × 10^9^/L, corticosteroid treatment, graft versus host disease, azotemia, prior surgery, prophylaxis, parenteral nutrition, antibiotic use in the last two weeks, length of hospitalization, CVC, sepsis), clinical course (sepsis, metastatic disease, complicated candidemia, transport to intensive care unit) and outcome were recorded. Moreover, mycological data, antifungal treatment and duration of treatment, laboratory findings at diagnosis and end of treatment were collected. Outcome was defined as survival or mortality at 28 days and at last follow-up.

Statistical analysis was performed using SPSS v.23 (IBM SPSS Statistics for Windows, Version 23.0. IBM Corp.: Armonk, NY, USA).

### Ethics Statement

The authors confirm that the ethical policies of the journal have been adhered to. Informed consent was obtained from legal guardians of participants at the time of malignancy diagnosis for the use of patients’ data in future studies. Moreover, approval for the study was received from the Ethical Review Board of the Aristotle University of Thessaloniki, Greece (57393/10 June 2016).

## 3. Results

From January 2015 to December 2019 19 episodes of candidemia occurred in 19 patients. Demographic characteristics and clinical data of the patients are shown in Table 1 and Table 2. The study population consisted of 19 patients, of which seven (36.8%) were female, and the mean age was 8.7 years (range 8 months–17 years). Candidemia was reported only in one infant (5.2%) in our study. The most common underlying malignancy was acute lymphoblastic leukemia (47.4%), followed by acute myeloid leukemia (21.2%). Moreover, the most frequent risk factors associated with candidemia observed in our patients were prolonged duration of hospitalization (30 days, range 1–141), presence of a central venous catheter (73.7%), antibiotics use (84.2%) as well as neutropenia during the last two weeks (56.3%). Finally, 21.1% of the patients received antifungal prophylaxis, with fluconazole being the most commonly prescribed agent (75%).

*Candida parapsilosis* was the most common species isolated from blood cultures accounting for 42.1%, followed by *C. albicans* (26.3%) and *C. famata* (15.8%) (Table 3). Other *Candida* species reported in our study were *C. krusei* (10.5%) and *C. tropicalis* (5.3%). No repeated episodes of candidemia in the same patient were recorded. In five patients (26.3%) simultaneous microbial growth was also detected (*C. albicans* and *Achromobacter*, *C. albicans* and *Pseudomonas*, *C. famata* and *S. epidermidis*, *C. parpsilosis* and *S. hominis*, *C. krusei* and *Trichosporon*).

The majority of the patients (84.2%) received antifungal monotherapy, with L-AmB being the most commonly used agent, followed by micafungin and fluconazole. The first-line antifungal agent was L-AmB in 9 cases (47.4%), echinocandins in 6 cases (31.6%) and azoles in 4 cases (21%). The CVC was removed in 78.6% of the patients after a median of 3 days (range 1–9) following the first positive blood culture. A new CVC was placed in 54.5% of the patients; the median time between the removal of the catheter and the placement of a new one was 13 days (range 0–32).

Complications occurred in seven patients (36.8%); organ dysfunction and sepsis were most frequently reported. The course of the disease was complicated only in two out of the eight patients with *C. parapsilosis* (25%) compared to five out of 11 patients (45.5%) with other *Candida* species (*p* > 0.05). In addition, mortality at 28 days was 26.3% (5/19) and overall 90-day mortality was 31.6% (6/19). Mortality according to *Candida* species was 60% (3/5) for *C. albicans*, 66.7% (2/3) for *C. famata* and 12.5% (1/8) for *C. parapsilosis*. Finally, CVC was removed in 33.3% (2/6) of the patients who died in comparison to 69.2% (9/13) of those who survived (*p* > 0.05).

## 4. Discussion

Candidemia is a major cause of morbidity and mortality worldwide, affecting primarily immunocompromised and hospitalized patients as children with hematological malignancies and solid tumors and those undergoing stem cell transplantation [1,2,3,5]. This national retrospective multicenter study provides valuable insights into the epidemiology and mortality of candidemia in children with malignancies in Greece. Our main findings were the predominance of *C. parapsilosis* and the mortality rate, which was higher than that in other pediatric studies. The majority of the patients (78.9%) had at least one risk factor for candidemia. The median length of hospitalization was 30 days and a CVC was present in 73.7% of the patients. Additionally, the patients were characterized by prior antibiotic use (84.2%) and neutropenia (56.3%) during the last two weeks.

In our study, *C. parapsilosis* was the most common (42.1%) causative agent of candidemia. In total, non-albicans species were identified in 73.7% of the cases. This comes in accordance with the global shift towards non-albicans species, mainly observed in hematology/oncology departments [2,17]. Similarly to our findings, Gamaletsou et al. reported a prevalence of 87.5% non-albicans candidemia in Greek adult hematology patients in a prospective multicenter study conducted from January 2009 to February 2012. They detected 40 episodes of candidemia in 40 patients. The most common species in their study was again *C. parapsilosis* (50%), followed by *C. tropicalis* (15%) and *C. albicans* (12.5%) [18]. Mesini et al. studied the incidence and mortality of *Candida* infections in an Italian pediatric tertiary care hospital. They also observed a prevalence of non-albicans species (66.7%) in oncology patients [19]. Moreover, this trend is also noted in a recent Australian retrospective 10-year study (2004–2013) by Bartlett et al. They retrospectively collected data from four tertiary pediatric oncology and hematopoietic stem cell transplant units. Of the 84 *Candida* species identified in their study, 73.8% were non-albicans—a percentage almost identical to ours—with *C. parapsilosis* being the most frequently isolated *Candida* (33.3%) [20]. Tragiannidis et al. documented a slight predominance of non-albicans candidemia (54.3%) in a German University Children’s Hospital, although *C. albicans* was the most common species (45.7%) [21]. In addition, Pemán et al. conducted a prospective, hospital population-based study in 30 Spanish hospitals over 13 months (from January 2009 to February 2010). They recorded 203 episodes of candidemia in 200 patients younger than 16 years of age, that were hospitalized in pediatric and neonatal intensive care units as well as in general wards (including hematology/oncology departments). Non-albicans species prevailed (63.5%) and *C. parapsilosis* was identified in the majority of the cases (46.8%) [22]. In Table 4, we summarize the aforementioned *Candida* epidemiology studies. Meanwhile, Nawrot et al. conducted a retrospective multicenter study between January 2006 and December 2007 in Poland. Non-albicans candidemia was reported in 78% of the cases in adult hematology patients, while *C. krusei* was the most frequently isolated species (24%) [23]. Finally, Tadec et al. recorded 36 episodes of candidemia in adult hematology patients within the framework of a retrospective study between 2004 to 2010. They likewise observed a prevalence of non-albicans species (88.9%) in adult hematology patients with *C. tropicalis* being the leading cause of candidemia in this sub-group of patients (27.7%) [24]. On the other hand, *C. albicans* still remains the most common species in Northern Europe [14,25], Middle East [26] and Asia [9,27]. However, the proportion of candidemia caused by *C. albicans* has lately decreased in Northern Europe [11] and U.S.A. [28].

This shift towards non-albicans species can be attributed partly to antifungal prophylaxis, as *C. glabrata* and *C. parapsilosis* show diminished susceptibility to azoles and echinocandins, respectively [29]. However, this is not adequately supported by the study of Gamaletsou et al. in Greek adult hematology patients. They reported that half of *C. glabrata* and 84.6% of *C. parapsilosis* were susceptible to antifungal agents [18]. Furthermore, the hands of health care workers play a crucial role. They are a common source of exogenous *Candida* species, which can infect the patients’ skin or CVCs and later cause catheter-related candidemia [8,29]. *Candida parapsilosis* exhibits strong affinity for foreign material and a growth advantage in total parenteral nutrition solution [12]. As shown by Lee et al., prolonged use of a CVC is associated with non-albicans candidemia [30].

Regarding the treatment of candidemia, 84.2% of the patients in the present study received monotherapy according to international guidelines. Removal of the CVC whenever possible was prompt after only a median of 3 days compared with 5.66 days reported in literature [21].

Mortality at 28 days was 26.3%, which is higher than the mortality reported in an Australian multicenter study (15.3%) with a population comparable to ours [20]. In a recent large retrospective multicenter European study of pediatric and neonatal candidemia, overall mortality at 30 days was 14.4%. In this study by Warris et al., mortality rate varied between groups. It ranged from 1.8% for patients from pediatric surgical wards to 18.3% and 27.8% for patients admitted to neonatal and pediatric intensive care units, respectively [17]. Tragiannidis et al. reported a mortality rate of 11.4% for pediatric patients. They included patients from four departments (pediatrics, cardiology, hematology/oncology, and surgery), but made no distinction between the groups when calculating the mortality rate [21]. An explanation for the high mortality in our study, which is comparable to that for patients in pediatric intensive care units, may lie in the fact that our patients were more severely ill. Two patients (10.5%) in our study were admitted to the pediatric intensive care unit due to complications of candidemia and they both died. In addition, the Stem Cell Transplant Unit recorded four cases of candidemia, half of which resulted in death (these cases do not overlap with the ones admitted to the intensive care unit). Moreover, Gamaletsou et al. [18] reported higher mortality rate in Greek adult hematology patients (45%) compared to other European studies (40.67% and 30.3%) [23,24]. The limited number of candidemia cases in combination with the higher mortality rate in the Greek population could be attributed to undiagnosed cases, which are often empirically treated. This should be further evaluated at a national level. Finally, Tsai et al. recorded 319 episodes of candidemia in 262 pediatric patients in a single center in Taiwan between 2003 and 2015. The mortality rate at 30 days was 25.2%, which is analogous to ours even though their study population was not limited to hematology/oncology patients. They also performed multivariate analysis and found that underlying hematological/oncological malignancy, delayed catheter removal, breakthrough candidemia and septic shock at onset were independently associated with candidemia attributable mortality [27].

## 5. Conclusions

*Candida* remains an important cause of bloodstream infections leading to significant morbidity and mortality. Non-albicans species have superseded *C. albicans* as the causative agent of candidemia in many countries. This global trend can be associated with antifungal prophylaxis as well as skin and CVC colonization probably. *Candida parapsilosis* is the most common Candida species isolated from blood cultures of Greek pediatric hematology/oncology patients, followed by *C. albicans*. Prevention, timely initiation of antifungal treatment and CVC removal are of paramount importance to survival and avoidance of complications.

## Figures and Tables

**Table 1 jof-06-00276-t001:** Demographic data, therapy and outcome of patients with candidemia.

	Mean (Range) or n/x (%)
Age (years)	8.7 (8 months–17 years)
Infants	1 (5.2%)
Girls	7 (36.8%)
**Department**	
Hematology-Oncology Unit	15 (78.9%)
Stem Cell Transplant Unit	4 (21.2%)
**Underlying disease**	
Acute lymphoblastic leukemia	9 (47.4%)
Acute myeloid leukemia	4 (21.2%)
Other malignancies	6 (31.6%)
**Therapy**	
Antifungal monotherapy	16 (84.2%)
Removal of central venous catheter	11/14 (78.6%)
**Outcome**	
Complications	7 (36.8%)
Mortality at 28 days	5 (26.3%)

**Table 2 jof-06-00276-t002:** Clinical characteristics and risk factors of patients with candidemia.

	Mean (Range) or n/x (%)
Length of hospitalization (days)	30 ^†^ (1–141)
Prolonged hospitalization (≥20 days)	11/19 (57.9%)
Central venous catheter	14 (73.7%)
Neutropenia in the last 2 weeks	9/16 (56.3%)
Neutropenia at diagnosis	7/17 (41.2%)
White blood cell count at diagnosis (K/μL)	2.92 ^†^ (0–18.33)
Absolute neutrophil count at diagnosis (K/μL)	2.31 ^†^ (0–15.4)
Antibiotics in the last 2 weeks	16 (84.2%)
Steroids in the last 2 weeks	8 (47.1%)
Prophylaxis	4 (21.1%)
Parenteral nutrition in the last 2 weeks	1/16 (6.3%)
Graft versus host disease	3 (17.6%)
Surgery in the last 2 weeks	2 (11.8%)

^†^ Median.

**Table 3 jof-06-00276-t003:** Distribution of Candida species.

*Candida* Species	No. of Episodes (%)
*C. albicans*	5 (26.3%)
Non-albicans	14 (73.7%)
*C. parapsilosis*	8 (42.1%)
*C. famata*	3 (15.8%)
*C. krusei*	2 (10.5%)
*C. tropicalis*	1 (5.3%)

**Table 4 jof-06-00276-t004:** Comparative table of Candida epidemiology studies.

	Our Study	Gamaletsou et al. [18]	Messini et al. [19]	Bartlett et al. [20]	Tragiannidis et al. [21]	Pemán et al. [22]
Year	2015–2019	2009–2012	2005–2015	2004–2013	1998–2008	2009–2010
Study population	Children	Adults	Children	Children	Children	Children
Number of participating centres	8	9	1	4	1	30
Department	Oncology, BMTU	Hematology, BMTU	Oncology	Oncology, BMTU	Pediatric, Oncology	Pediatric, PICU, NICU
Country	Greece	Greece	Genova, Italy	Australia	Münster, Germany	Spain
Total No of cases	19	40	27	84	35	203
*C. albicans*	5 (26.3%)	5 (12.5%)	9 (33.3%)	22 (26.2%)	16 (45.7%)	74 (36.5%)
Non-albicans	14 (73.7%)	35 (87.5%)	18 (66.7%)	62 (73.8%)	19 (54.3%)	129 (63.5%)
*C. parapsilosis*	8 (42.1%)	20 (50%)	-	28 (33.3%)	6 (17.1%)	95 (46.8%)
*C. famata*	3 (15.8%)	-	-	-	-	1 (0.5%)
*C. krusei*	2 (10.5%)	-	-	13 (15.5%)	-	2 (1%)
*C. tropicalis*	1 (5.3%)	6 (15%)	-	3 (3.6%)	2 (5.7%)	12 (5.9%)
*C. glabrata*	-	4 (10%)	-	7 (8.3%)	5 (14.3%)	8 (3.9%)
Other *Candida* species	-	5 (12.5%)	-	11 (13.1%)	6 (17.1%)	11 (5.4%)
Mortality	26.3%	45%	-	15.3%	11.4%	-

BMTU: Bone Marrow Transplant Unit, PICU: Pediatric Intensive Care Unit, NICU: Neonatal Intensive Care Unit.

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
