# Peer review of "Candidemia in Children with Malignancies: Report from the Infection Working Group of the Hellenic Society of Pediatric Hematology-Oncology"

_jof, 2020, doi:10.3390/jof6040276_

Round 1
Reviewer 1 Report
The paper from E. Vasileiou et al. has been revised and I find this version somewhat improved from the original one. However, this is a small study comprising only 19 cases of candidemia. Although candidemia is rare in children, the number is so small that statistical analyses give little meaning. Therefor the paper needs to focus on its 2 main finding – C. parapsilosis is the most common species found and the mortality is in the high end of what has been reported in children. I suggest particularly the Discussion needs to be rewritten and suggest the following as practical advice:
- The Discussion is too long. The first paragraph should be omitted.
- The second paragraph is a repetition of Results and should either be omitted or one could mention the potential important findings of this study; C. parapsilosis and mortality -
- as an introduction to the Discussion.
- In the third paragraph, the fact that C. parapsilosis being dominating is partly discussed. Firstly, it states that others also how found increases in non-albicans and an increase in Secondly, in the fourth paragraph, the discussion on why there is a shift in non-albicans and an increase in C. parapsilosis takes place. Increased resistance is suggested as a possibility, though there is no reference to other epidemiological studies that includes susceptibility testing from Greece. And susceptibility data on these 19 isolates are for some reason not included. In many countries, C. parapsilosis is not associated with resistance, not even against echinocandins even though the break points for these drugs are higher. Clinically, it seems to have little relevance. In the Conclusion, poor hygiene is suggested as an explanation, but is this so much worse in Greece than in other countries? I know it is common that families takes care of their hospitalized relatives to a much greater degree than in e.g. western Europe, but surely not in these dramatically sick patients? If so, it should be noted.
- C. parapsilosis. However, the paragraph is too long, it needs to be focused and in particularly on children. Although the fact that the species is also common in adults may be important. Table 4 should only include pediatric studies.
- Paragraph 5 is again a summary of Results and should be omitted. Again if you say something about therapy, we need to know susceptibility, which is for some reason lacking.
- In the sixth paragraph, mortality is discussed. Interestingly, it is in the higher numbers of what has previously been reported. But that is also the fact among greek adults. The risk factors seem to be the same. But we do not know if resistance to therapy is part of the problem. This must be included if you are make serious effort to go into this matter.
- In the Conclusion: It is stated in sentence 222 and 223 that C. parapsilosis is the most common species found in blood cultures in Greek pediatric hemato/oncology patients. Should it be “the most common Candida species” or is this actually the case?
Author Response
We would like to thank both Reviewers for assessing our study and offering constructive criticism. In the revised manuscript, we have addressed the Reviewers’ comments.
Response to Reviewer #1
- The Discussion is too long. The first paragraph should be omitted.
- The second paragraph is a repetition of Results and should either be omitted or one could mention the potential important findings of this study; C. parapsilosis and mortality - as an introduction to the Discussion.
First paragraph was reduced significantly in size (deleted lines 132-138) and was merged with the second paragraph. A sentence stating the main findings was added, as suggested (lines 140-142). Part of the second paragraph was also deleted (lines 145-146).
- In the third paragraph, the fact that C. parapsilosis being dominating is partly discussed. Firstly, it states that others also how found increases in non-albicans and an increase in Secondly, in the fourth paragraph, the discussion on why there is a shift in non-albicans and an increase in C. parapsilosis takes place. Increased resistance is suggested as a possibility, though there is no reference to other epidemiological studies that includes susceptibility testing from Greece. And susceptibility data on these 19 isolates are for some reason not included. In many countries, C. parapsilosis is not associated with resistance, not even against echinocandins even though the break points for these drugs are higher. Clinically, it seems to have little relevance.
The findings of the study by Gamalatsou et al. regarding susceptibility of non-albicans species in Greece were added (lines 185-187). Unfortunately, we have no susceptibility data for all of our patients and due to this we prefer not to include data for a study with 8 cases of C. parapsilosis candidemia. The scope of this study focuses on the epidemiology of candidiasis, so data regarding susceptibilities were not included.
- In the Conclusion, poor hygiene is suggested as an explanation, but is this so much worse in Greece than in other countries? I know it is common that families takes care of their hospitalized relatives to a much greater degree than in e.g. western Europe, but surely not in these dramatically sick patients? If so, it should be noted.
The phrase “due to poor conformity with good hygiene practices” (line 225) was deleted. This is suggested in literature, but we have not collected data to support it.
- parapsilosis. However, the paragraph is too long, it needs to be focused and in particularly on children. Although the fact that the species is also common in adults may be important. Table 4 should only include pediatric studies.
The last two columns (adults) of Table 4 were deleted. The second column summarizing the results of a Greek study in adult hematology patients was kept for comparative reasons, because it is the only study that discusses the epidemiology of candidemia in Greece. A sentence was moved from lines 175-176 to 167-168 because of this deletion. Changes were made in lines 169-171 in order to reduce the size of the paragraph. Moreover, changes were also made in lines 211-212 (the mortality rates were added in parenthesis and the reference to Table 4 was deleted).
- Paragraph 5 is again a summary of Results and should be omitted. Again if you say something about therapy, we need to know susceptibility, which is for some reason lacking.
Lines 193-194 and 195-196 were deleted. The sentences commenting that international treatment guidelines were followed and comparing the time of CVC removal with literature were kept.
- In the sixth paragraph, mortality is discussed. Interestingly, it is in the higher numbers of what has previously been reported. But that is also the fact among greek adults. The risk factors seem to be the same. But we do not know if resistance to therapy is part of the problem. This must be included if you are make serious effort to go into this matter.
As mentioned previously, no susceptibility data were collected for our patients. This study was focused on the epidemiology of candidemia in pediatric hematology oncology patients.
- In the Conclusion: It is stated in sentence 222 and 223 that C. parapsilosis is the most common species found in blood cultures in Greek pediatric hemato/oncology patients. Should it be “the most common Candida species” or is this actually the case?
The term “Candida” was added in line 226 to avoid further confusion (revised manuscript line 213).
Reviewer 2 Report
None
Author Response
We would like to thank both Reviewers for assessing our study and offering constructive criticism. In the revised manuscript, we have addressed the Reviewers’ comments.
Response to Reviewer #2
No comments were reported by reviewer 2.
This manuscript is a resubmission of an earlier submission. The following is a list of the peer review reports and author responses from that submission.
Round 1
Reviewer 1 Report
E. Vasileiou and coworkers have conducted a retrospective study including patients from pediatric hemato-oncologic wards of 8 medical centers in Greece in the period 2015-219. They report the findings of 19 patients.
My main objection to this study is that it is too small to have any general interest. Their findings of 1) a very low number of candidemi in their population, 2) a majority of C. parapsilosis and 3) a rather high mortality rate are not properly discussed.
According to the study by Warris et al., the pediatric hemato-oncological population comprised 16,9% of the total population which should give about 10 patients per study site. The number is much lower in these 8 centers in Greece. Why is that?
The finding of higher incidence of C. parapsilosis is typical in neonates and in southern Europe. But there are no neonates in this study. Is this a reflection of a southern European situation? Did these isolates have any susceptibility issues? None are given.
The high mortality rates are also not properly discussed. They are almost double of what others report.
Furthermore, the study are lacking in details one would expect in such a study:
- Apart from the range of age – how many were infants, how many were children < 1 year?
- Obviously the greatest risk factor for candidemia in these patients were their underlying hematologic malignant disease. This seems to have been forgotten. Other risk factors come with the underlying disease.
- Under risk factors there is a mixture of data that also includes underlying disease e.g. zotemia.
- In table 1 demographics and clinical characteristics are given – and then some more. I miss a proper setup: 1) underlying disease, 2) risk factors 3) outcome. It is all mixed up.
- 7 patients received prophylaxis and still got their candidemias. Why is that? There is a mention of maybe too low dosage, but were the Candida sp. susceptible to the given prophylaxis?
- How many patients were treated with liposomal AMF B, echinocandin or fluconazole?
- There are a few comments that are incorrect:
- in sentence 46: there are hundreds of Candida species – well there is about 160 of them.
- in s. 48: C. auris is common – no, it is fortunately not.
Reviewer 2 Report
This paper presents a concise report on the burden of candidaemia among children with malignancies from 7 centers in Greece.
Major comments
The denominator for this study is not known or stated. So we are not sure whether candidaemia is common in this setting or not.
It would have been appropriate for the authors to compare those who developed the disease and those who didn't, again this data is lacking. Any reason for this ?
Minor comments
1- Authors should list the centres
2- Ethical statement - was ethical clearance received from the appropriate IRB? How dis the legal guardians/parents consent ? retrospectively ? this has to be clearly stated.